# Leveraging Multimodal LLM Descriptions of Activity for Explainable Semi-Supervised Video Anomaly Detection

**Furkan Mumcu**[*]                                                    *furkan@usf.edu*
*Department of Electrical Engineering*
*University of South Florida*

**Michael J. Jones**                                                   *mjones@merl.com*
*Mitsubishi Electric Research Laboratories (MERL)*

**Anoop Cherian**                                                      *cherian@merl.com*
*Mitsubishi Electric Research Laboratories (MERL)*

**Yasin Yilmaz**                                                       *yasiny@usf.edu*
*Department of Electrical Engineering*
*University of South Florida*

**Reviewed on OpenReview:** *https://openreview.net/forum?id=dfc2HpDSlH*

## Abstract

Existing semi-supervised video anomaly detection (VAD) methods often struggle with detecting complex anomalies involving object interactions and generally lack explainability. To overcome these limitations, we propose a novel VAD framework leveraging Multimodal Large Language Models (MLLMs). Unlike previous MLLM-based approaches that make direct anomaly judgments at the frame level, our method focuses on extracting and interpreting object activity and interactions over time. By querying an MLLM with visual inputs of object pairs at different moments, we generate textual descriptions of the activity and interactions from nominal videos. These textual descriptions serve as a high-level representation of the activity and interactions of objects in a video. They are used to detect anomalies during test time by comparing them to textual descriptions found in nominal training videos. Our approach inherently provides explainability and can be combined with many traditional VAD methods to further enhance their interpretability. Extensive experiments on benchmark datasets demonstrate that our method not only detects complex interaction-based anomalies effectively but also achieves state-of-the-art performance on datasets without interaction anomalies.

## 1 Introduction

Video anomaly detection (VAD) has emerged as an active research field due to its significant applications in security and public safety. The rapid growth of surveillance video data makes it impractical for humans to monitor thoroughly. Consequently, VAD algorithms play a vital role in identifying unusual events in video footage, enabling human operators to focus their attention on reviewing these flagged incidents. In this paper, we focus on the formulation in which nominal videos (also called training videos) containing only normal activities in a particular scene are provided for learning a model. The goal is to identify when and where anomalous activities happen within test videos of the same scene. Along with other papers (Baradaran & Bergevin, 2024; Wu et al., 2024a), we refer to this version of the problem as *semi-supervised* VAD because of the availability of normal, but not anomalous, training video. In Table 1, we summarize the properties of some recent papers in this area to highlight the differences between our work and other recent approaches.

---

[*]Furkan Mumcu did part of this work as an intern at MERL

Existing semi-supervised video anomaly detection methods suffer from several weaknesses. Recent studies struggle to provide explainability for detected anomalies. While some existing work (Singh et al., 2023; 2024) explored more explainable approaches by interpreting the input features, they do not offer direct, textual explanations. Furthermore, the ComplexVAD dataset (Mumcu et al., 2025b) introduced interaction-based anomalies and demonstrated that detecting these anomalies is even more challenging than identifying the simpler anomalies presented in earlier datasets. Hence, we propose a new method to address the drawbacks present in existing works, including the lack of focus on complex anomalies and explainability. By utilizing a Large Language Model-based detection approach, we introduce an effective video anomaly detection framework that excels at identifying complex anomalies and features built-in explainability.

Large Language Models, especially in their multimodal forms (MLLMs), have demonstrated strong capabilities in interpreting and accurately describing complex situations from both text and images. GPT-4 with vision (Achiam et al., 2023), for example, can analyze visual inputs and produce detailed, contextually relevant descriptions. Existing work has explored the use of MLLMs for video anomaly detection. Most of these methods are designed for a multi-scene, weakly supervised setting. While such approaches are effective for detecting a common set of anomalies across multiple scenes, they fall short in identifying scene-specific anomalies, which are central to the semi-supervised, single-scene VAD task. To address this limitation, we propose a novel MLLM-based framework that: (1) considers object interactions in addition to individual object activities, (2) leverages an MLLM to generate high-level descriptions used in an exemplar-based model that can be easily extended without retraining deep networks for each new scene, and (3) enables both spatial and temporal localization of anomalies.

In our method, we adopt an object-level feature extraction strategy similar to (Mumcu et al., 2025b), using an object detector and tracker. We then employ an MLLM agent to describe object activities and interactions by querying it with pairs of frames, separated in time by a fixed interval, to capture temporal information. A key novelty is using these MLLM-generated textual descriptions directly as features for modeling normal activity. By removing redundant descriptions (Section 3), we construct a final set of textual exemplars representing nominal videos. This exemplar set is then used to detect anomalies: descriptions from anomalous interactions are expected to deviate from nominal ones. In summary, our contributions are:

- We introduce the first MLLM-based approach for video anomaly detection specifically designed to identify complex anomalies caused by object interactions.

- We propose a novel use of MLLMs for VAD: instead of directly deciding the presence of an anomaly, as done in prior work, our method first derives a representation (using MLLM descriptions) of what is normal in a scene and then identifies anomalies based on deviations from this normal representation.

- We demonstrate that our method offers built-in explainability. Furthermore, we show that it can be integrated with many existing VAD approaches to enhance their interpretability.

- We evaluate our method on multiple benchmark datasets and show that it achieves state-of-the-art performance.

## 2 Related Work

Many different formulations of the video anomaly detection (VAD) problem have been studied in recent years: semi-supervised (Hasan et al., 2016; Ionescu et al., 2019; Liu et al., 2018; Wang et al., 2021), weakly supervised (Sultani et al., 2018), unsupervised (Zaheer et al., 2022), training-free (Zanella et al., 2024), single-scene (Ramachandra et al., 2020b) and multi-scene (Luo et al., 2017). Each of these formulations has applicability in different scenarios. This paper focuses on the single-scene, semi-supervised version of video anomaly detection because it corresponds to the most common use case that we have observed in the real-world; namely, that of a surveillance camera at a particular location in the world in which video of normal activity is abundant but video containing anomalies is rare and expensive to collect. An important attribute of the single-scene VAD problem formulation is that anomalies are scene dependent, i.e., what is anomalous in one scene may be normal in another. This is in contrast to the multi-scene problem formulation in which

| Method | Focus | Features | | | Task Objective | |
|---|---|---|---|---|---|---|
| | | MLLM Usage | Interaction Anomalies | Explain-ability | Semi Supervised | Scene Specific |
| VadCLIP Wu et al. (2024d) | Using CLIP for multi-scene weakly supervised VAD | ◐ | ○ | ○ | ○ | ○ |
| Vera Ye et al. (2025) | Optimizing prompts for MLLM-based weakly supervised VAD | ● | ○ | ● | ○ | ○ |
| Holmes-VAU Zhang et al. (2025) | MLLM finetuning for anomaly understanding across multi-scene | ● | ○ | ● | ○ | ○ |
| AnomalyRuler Wu et al. (2024c) | Scene-specific rule generation and frame-level descriptions via MLLMs | ● | ○ | ◐ | ● | ● |
| VLAVAD Li & Jiang (2024) | Using frame level MLLM descriptions for semi-supervised VAD | ● | ○ | ○ | ● | ● |
| EVAL Singh et al. (2023) | Using object-level features for explainable VAD | ○ | ○ | ◐ | ● | ● |
| T-EVAL Singh et al. (2024) | Object-level, tracklet-based explainable VAD | ○ | ○ | ◐ | ● | ● |
| Scene-Graph (Mumcu et al., 2025b) | Exemplar and scene graph for interaction-based VAD | ○ | ● | ◐ | ● | ● |
| Ours | Object-level MLLM descriptions for interaction-based VAD | ● | ● | ● | ● | ● |

Table 1: Summary of properties and differences of some recent work in VAD. Our work is the only one designed for semi-supervised, scene-specific VAD that handles interaction anomalies and provides textual explanations of anomalies.

anomaly types are common across different scenes. Finally, the single-scene VAD formulation typically contains anomalies that are location-dependent (e.g. a pedestrian may be anomalous in some locations but not others). Such location-dependent anomalies do not occur in multi-scene VAD because they require scene-specific models. These differences imply that the multi-scene version of VAD is not a generalization of single-scene VAD (Ramachandra et al., 2020b). Solutions for one version generally do not work well on the other version.

Early work on the single-scene, semi-supervised version of VAD includes many papers based on frame reconstruction (Hasan et al., 2016; Ionescu et al., 2019) or frame prediction (Liu et al., 2018; Wang et al., 2021). One big disadvantage of such approaches is that they require training a deep network on video from each new scene. To avoid this disadvantage, others (Doshi & Yilmaz, 2020b;a; Ramachandra et al., 2020a; Ramachandra & Jones, 2020; Singh et al., 2023; 2024; Mumcu et al., 2025b) used an exemplar-based model that extracts a set of representative features from normal video but does not require any deep network training for each scene. We also use this idea in our work.

## 2.1 MLLMs in Video Anomaly Detection

Recently, researchers have explored ways of taking advantage of large language models or vision language models for VAD. These approaches, however, vary significantly depending on the VAD problem formulation.

**Weakly-Supervised and Training-Free VAD.** Many approaches to VAD that use an MLLM try to use its built-in knowledge of what is normal instead of learning normality from normal training video (Zanella et al., 2024; Ye et al., 2025; Lv & Sun, 2024). A disadvantage of such approaches is that they cannot handle scenarios such as a boxing gym in which activity that is typically rare and anomalous (fighting) is actually very common. Another important difference between our work and most other MLLM-based approaches to VAD is that others have focused on the weakly supervised, multi-scene version of VAD (Li et al., 2025; Wu et al., 2024b; Zanella et al., 2024; Ye et al., 2025; Lv & Sun, 2024).

**Semi-Supervised VAD.** Compared to the weakly-supervised setting, the capabilities of MLLMs are understudied in the semi-supervised setting. Wu et al. (2024c) addresses semi-supervised, single-scene VAD, but with a fundamentally different methodology. Their approach uses an MLLM to first generate a set of textual rules from normal video and then applies these rules during inference. In contrast, our method directly compares MLLM-generated text descriptions between normal and test videos without an intermediate rule-generation step. Furthermore, their model processes full video frames, whereas we employ an object-centric approach that analyzes regions around objects and their pairs. This key design choice allows our method to model object interactions and perform both spatial and temporal localization, while their approach is limited to temporal localization only.

Li & Jiang (2024) also focuses on semi-supervised single scene anomaly detection and proposes an MLLM-based method. Like our method, they first detect and track objects and then query an MLLM to yield high-level descriptions of objects and their activity. However, they do not model object interactions, and do not use an exemplar-based model.

In real-world scenarios, many anomalies are characterized by an unusual interaction between two objects. Therefore, one of the important aspects of our method is its modeling of object interactions in the scene to identify anomalous interactions. There have been a few other methods that also modeled interactions (Chen et al., 2018; Doshi & Yilmaz, 2023; Sun et al., 2020; Szymanowicz et al., 2021; Mumcu et al., 2025b). While there are many differences among these methods, one of the main differences compared to our current work is that none of them take advantage of MLLMs. In our approach, the textual descriptions of object activity and interactions provided by an MLLM yield powerful cues for a high-level understanding of a scene. Language provides a natural abstraction for compositional structure and relational semantics (e.g., interactions, roles, and intent), which are difficult to capture reliably with low-level features alone. In the one-class setting, this abstraction enables a compact and human-aligned representation of normal behavior, where deviations correspond to semantically meaningful changes rather than pixel-level noise.

## 3 Our Approach

We propose a novel method to detect anomalies in video. The pipeline of our method (which we call MLLM-EVAD, for MLLM-based Explainable Video Anomaly Detection) is depicted in Figure 1. The main idea is to use a set of textual descriptions of the activity seen in normal training video of a scene as the model of normality. These textual descriptions are obtained using an MLLM. Anomalies in testing video are found by comparing textual descriptions of the activity in testing frames with the set of normal descriptions. A test description that is dissimilar from all normal descriptions indicates an anomaly. The initial step of our method is detecting and tracking objects followed by identifying pairs of objects that are in close spatial proximity and therefore likely to be interacting. Then, for each frame and a future frame offset by a fixed time interval, we generate crops around pairs of interacting objects as well as single objects that are not close to other objects. The crops are given to an MLLM to obtain natural language descriptions of objects' activities. These descriptions are embedded into a vector space using a sentence embedding model. By applying an exemplar selection algorithm to all of the sentence embeddings found in nominal training video, we construct representative sets of embeddings for both object pairs and single objects. Anomaly detection is performed by comparing sentence embeddings extracted from testing frames to the exemplar sets, and assigning anomaly scores based on their distance to the most similar exemplar.

### 3.1 Object Detection and Tracking

In our proposed method, we want to build a model of what objects do and how they interact in video containing normal activity. This requires detecting objects and tracking them across frames in training videos. We also find pairs of objects that are likely to be interacting according to their spatial proximity.

A video $V$ is a collection of $M$ frames $\{F_i\}_{i=1}^{M}$, such that $V = [F_1, F_2, ..., F_M]$. Each frame $F_i$ is sent to the object detector $O$, which returns $X$ number of detected objects. The object detector provides, for each object $o$, the location $l = (x_o, y_o)$ which is the $x$ and $y$ coordinates of the center of the object, $b = (w_o, h_o)$ which is the width and height of the bounding box for the object, and class id $c$. The output of the object detector is then $O(F) = [o_1, o_2, ..., o_X]$, where each object $o_i$ is represented by $o_i = [b, c, l]$. After detecting objects in a frame, they are then tracked using an object tracker. Each detected object $o$ is sent to the object tracker, which returns $x$ and $y$ coordinates for that object in the subsequent frames. In our method, we track objects for 30 frames. Therefore, for every object, we acquire the trajectory $\theta = \{(x_1, y_1), (x_2, y_2), ..., (x_{30}, y_{30})\}$.

Next, we pair objects according to their spatial proximity in the frame. To determine which objects should be paired, we need to calculate the 3D distances between each pair of objects. To calculate the 3D distance we need to derive the 3D coordinates of the object locations by estimating a pseudo-depth since we do not have access to actual depth measurements. Given two objects $o_1$ and $o_2$, we have 2D coordinates $l_1 = (x_1, y_1)$ and $l_2 = (x_2, y_2)$. Then we define a relative depth, $z$, between two objects by taking the absolute difference of $y$ values such that $z = |y_1 - y_2|$. This estimate of pseudo-depth assumes that objects are resting on the ground plane and the ground plane is farther from the camera the closer it is to the top of the image. The 3D distance $d$ can then be calculated by taking the Euclidean distance between 3D coordinates $(x_1, y_1, z)$ and $(x_2, y_2, 0)$. Any objects that have a 3D distance $d$ smaller than a predetermined threshold $h$ are paired.

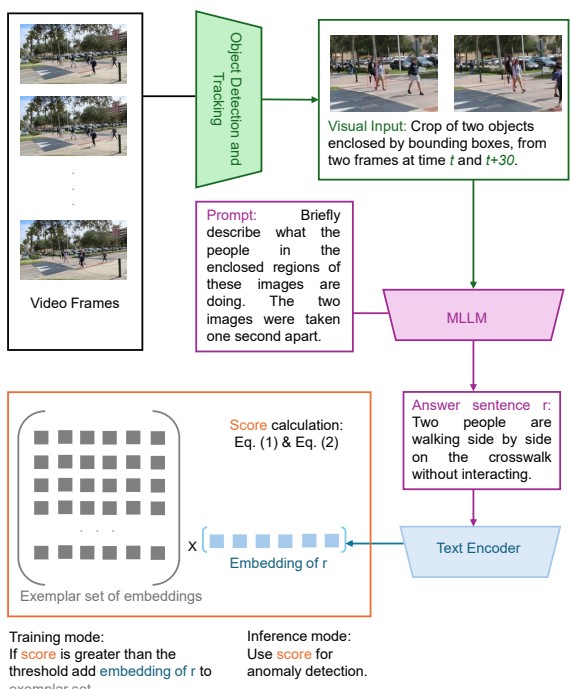

Figure 1: The pipeline of our method (MLLM-EVAD) for frame to textual description generation with the help of an object detector, object tracker and MLLM. Please see the text for more explanation.

Due to applying the threshold, not every single object is necessarily connected to another object, which leads to having single objects in addition to object pairs.

We use a temporal gap of approximately one second between the paired frames as a practical trade-off between capturing meaningful temporal changes and maintaining object identity consistency in a static-camera, single-scene setting. Shorter temporal gaps often result in nearly identical visual content and textual descriptions with limited motion cues, while significantly larger gaps increase the risk of object drift, missed interactions, or changes in object visibility. Importantly, the proposed framework is not tightly coupled to a specific temporal gap; the paired-frame formulation serves as a lightweight mechanism for incorporating local temporal context, and similar behavior can be captured with other reasonable temporal offsets depending on the scene dynamics.

## 3.2 Generating Textual Descriptions

At the end of the object detection and tracking stage, for each frame $F_t$, we obtain a set of object pairs and a set of single objects. For each of these, we generate two cropped regions that capture the relevant spatial area for both the current frame $F_t$ and a future frame $F_{t+\Delta}$ (e.g., $\Delta = 30$), assuming a static camera (as depicted in Figure 1). This ensures that both crops contain the same background and object regions, allowing for consistent comparisons across time. We provide the details of the cropping process in the Appendix. Red rectangles are drawn around each object in the cropped images using the bounding boxes detected by the object detector. The rectangles indicate to the MLLM which objects are being referred to in the prompt.

We then obtain a sentence $r$ that describes the interaction between objects $o_1$ and $o_2$ by querying an MLLM agent with the corresponding image crops $C_t$ and $C_{t+\Delta}$, along with the user prompt: *"Briefly describe what the <object name>s in the enclosed regions of these images are doing. The two images were taken one second apart."* For single-object cases, the prompt is adjusted to: *"Briefly describe what the <object name> in the enclosed region of these images is doing. The two images were taken one second apart."* In both prompts, the *<object name>* placeholder is replaced by the class label $c$ obtained during feature selection.

An additional "system" prompt is also used with the MLLM: *"You will be provided with two frames from a video and asked to describe what objects that are indicated by bounding boxes in the video are doing. Your task is to answer the query in a simple sentence. If there is any interaction between the indicated objects, a description of the interaction should be given"*

The MLLM-generated responses are then used in both the model building and anomaly detection stages.

We note that our use of image-based multimodal language models is a deliberate design choice. During development, we explored video-language models that process short clips or frame sequences; however, we observed that such models tend to produce coarse, scene-level descriptions that are less sensitive to fine-grained object interactions and localized activity changes. In contrast, querying an image-based MLLM with paired object-centric crops from temporally separated frames provides sufficient temporal context while preserving spatial grounding and object identity. This design aligns naturally with our goal of scene-specific modeling of normal behavior through localized object interactions in a semi-supervised setting, rather than general video understanding.

### 3.3 Model Building

We follow a similar exemplar selection strategy as introduced in recent works (Ramachandra & Jones, 2020; Singh et al., 2023; 2024; Mumcu et al., 2025b). For all frames in a video, we collect all textual descriptions generated from object pairs into one set and all textual descriptions generated from single objects into another set. Then for each of the sets we run an exemplar selection algorithm which selects a subset of the elements of the set such that no two members of the subset are near each other according to a distance function. The intuition behind exemplar selection is to simply remove redundant (or nearly redundant) elements from the set leaving behind a compact, representative subset of exemplars. Various distance functions between sentences can be used to compare textual descriptions. In most of our experiments we use Sentence-BERT (Reimers & Gurevych, 2019), but we also test using BLEU (Papineni et al., 2002) and METEOR (Banerjee & Lavie, 2005).

Given a set $S$, the exemplar selection algorithm proceeds as follows: (1) Initialize the exemplar set to NULL. (2) Add the first element of $S$ to the exemplar set. (3) For each subsequent element of $S$, find its distance to the nearest instance in the exemplar set. If this distance is above a threshold, $th$, then add the element to the exemplar set.

Since we obtain a textual description $r$ for each object pair or single object, we transform these descriptions into sentence embeddings using a Natural Language Processing (NLP) model $N$, such that $I_r = N(r)$. The distance $D$ between two textual descriptions, $r_1$ and $r_2$, is then computed as:

$$D(r_1, r_2) = 1 - \cos\left(I_{r_1}, I_{r_2}\right) \tag{1}$$

where $\cos(\cdot, \cdot)$ denotes the cosine similarity between the corresponding embeddings. Using the distance function $D$ and the described exemplar selection algorithm, we construct two separate exemplar sets: $E_{pair}$ and $E_{single}$, containing sentence embeddings of textual descriptions for object pairs and single objects, respectively.

### 3.4 Anomaly Detection in Test Video

After constructing the exemplar sets from nominal videos, the next step is to detect anomalies in a test video from the same scene. Similar to the model-building phase, we extract textual descriptions and encode them into sentence embeddings $I_r$. The anomaly score for a given object pair (or single object) is then computed based on its similarity to the corresponding exemplar set. Specifically, the score is defined as:

$$\text{score}(r) = 1 - \max\left(\cos\left(I_r, e\right)\right), \quad e \in E_{\text{pair}} \text{ or } E_{\text{single}} \tag{2}$$

depending on whether the input is an object pair or a single object. A higher score indicates a greater deviation from the nominal descriptions and is thus more likely to be anomalous.

### 3.5 Combination with Previous Methods

While the textual descriptions generated by an MLLM provide powerful high-level features for modeling the normal activity in a scene and for detecting anomalies in the scene, some anomalies require more detailed, fine-grained features for detection. For this reason, it can be advantageous to combine our method with previous methods that use more detailed features such as object sizes and trajectories. Combining with another method is especially easy to do if the other method is also object-based and uses an exemplar-based model.

In our experiments, we show results when combining our method separately with two other methods; namely, the scene-graph method of Mumcu et al. (2025b) and the tracklet method of Singh et al. (2024).

Our method of modeling interactions was inspired by the scene-graph method of Mumcu et al. (2025b). Their method builds a scene graph for each frame where each object is a node and objects that are nearby and, therefore possibly interacting, are connected by an edge. The attributes stored at each node are the object's location, size, object class, trajectory and skeletal pose. Then an exemplar-based model is created for every connected pair of nodes across all scene graphs in all nominal frames. A separate exemplar-based model is also created for all singleton nodes across all scene graphs in all nominal frames. To combine our method with theirs, we can simply add the MLLM-generated textual descriptions to each connected pair of nodes and to each singleton node. The distance function used to compare nodes takes the maximum over all attribute distances including the distance between textual descriptions.

In the case of the tracklet method of Singh et al. (2024), objects in each frame are also detected and tracked. Similar to Mumcu et al. (2025b), for each object, a set of attributes are stored including the location, size, object class and trajectory. An exemplar-based model is then learned from all of the objects found in the nominal training videos using a distance function for comparing objects that takes the maximum over each attribute distance. To combine our method with theirs, we again only need to add our MLLM-generated textual descriptions as another attribute associated with each object. In this case, object interactions are not modeled. We use this combination to test on datasets that do not contain anomalous object interactions.

## 4 Experiments

In our experiments, we use the following configurations:

**Datasets:** We evaluate MLLM-EVAD on three video anomaly detection benchmark datasets which are all publicly available under a CC-BY-SA-4.0 license. ComplexVAD (Mumcu et al., 2025b) is a large-scale dataset with a total of over 3.6 million frames, with 2,069,941 frames for training and 1,611,497 for testing. It features interaction-based anomalies and serves as our main focus, since our method leverages descriptions of object activity and interactions for video anomaly detection. The scene features a crosswalk, pedestrian sidewalks, and a two-lane street, which are heavily used. It demonstrates interaction anomalies such as a person leaving an object on the ground, a person sitting on a car, or a dog walking without a person or leash, where the individual objects are normal, but their interactions are anomalous.

In addition, we evaluate performance on the Avenue and Street Scene datasets. Avenue (Lu et al., 2013) contains short surveillance videos of a campus walkway, with anomalies such as running, throwing objects, or walking in the wrong direction. Street Scene (Ramachandra & Jones, 2020) includes longer videos of a city street environment, where anomalies involve vehicles or pedestrians behaving unusually, including cars driving on the wrong side of the road, pedestrians crossing at non-designated areas, or bikers riding on the sidewalk. Avenue contains 30,652 frames, split nearly evenly between 15,328 training frames and 15,324 testing frames. Street Scene is a larger dataset with a total of 203,257 frames, of which 56,847 are used for training and 146,410 for testing.

**Implementation Details:** We use Detectron2 (Wu et al., 2019) and ByteTrack (Zhang et al., 2022) as the object detector and object tracker in our method. Sentence-BERT (Reimers & Gurevych, 2019) (using the pretrained weights of *all-MiniLM-L6-v2* variant) is used as the text embedding model to encode activity descriptions. Initially, GPT-4o (Hurst et al., 2024) was used as the multi-modal MLLM agent. However, with the recent introduction of Gemma 3 (Team et al., 2025), we re-conducted the experiments

| Method | RBDC | TBDC | Frame |
|---|---|---|---|
| MemAE (Gong et al., 2019) | 0.05 | 0.0 | 58.0 |
| EVAL (Singh et al., 2023) | 10.0 | 62.0 | 54.0 |
| AnomalyRuler (Wu et al., 2024c) | N/A | N/A | 56.0 |
| Scene-Graph (Mumcu et al., 2025b) | 19.0 | 64.0 | 60.0 |
| MLLM-EVAD | **24.0** | **68.0** | **61.0** |
| Scene-Graph + MLLM-EVAD | **25.0** | **70.0** | **63.0** |

Table 2: Results on the ComplexVAD dataset. Each entry is the area under the curve (AUC) as a percentage for the three benchmark methods using the RBDC, TBDC and Frame-Level evaluation criteria. Bold face indicates the best score and blue indicates second best.

| Method | RBDC | TBDC | Frame |
|---|---|---|---|
| Auto-encoder (Hasan et al., 2016) | 0.3 | 2.0 | 61.0 |
| Dictionary method (Lu et al., 2013) | 1.6 | 10.0 | 48.0 |
| Flow baseline (Ramachandra & Jones, 2020) | 11.0 | 52.0 | 51.0 |
| FG Baseline (Ramachandra & Jones, 2020) | 21.0 | 53.0 | 61.0 |
| EVAL (Singh et al., 2023) | 24.3 | 64.5 | N/A |
| Contextual GMM (Yang & Radke, 2025) | **34.0** | 62.5 | **67.0** |
| T-EVAL (Singh et al., 2024) | 30.9 | **72.9** | 66.0 |
| T-EVAL + MLLM-EVAD | **31.1** | **73.5** | **66.8** |

Table 3: Results on the Street Scene dataset using the area under the curve (AUC) (as a percentage) for the RBDC and TBDC evaluation criteria for different methods. Bold face indicates the best score and blue indicates second best.

on ComplexVAD using Gemma 3. Hence, the results on ComplexVAD are reported with both Gemma 3 and GPT-4o, while the results on the other datasets are reported with GPT-4o. We choose a threshold $th = 0.65$ for exemplar selection, which results in a modest number of total exemplars while maintaining strong detection performance. As observed in prior exemplar-based VAD methods, the choice of $th$ primarily affects model size rather than test accuracy, with performance remaining stable across a reasonable range of threshold values (Singh et al., 2023; 2024). In our framework, exemplar selection serves to remove near-duplicate descriptions and construct a compact representation of normal behavior, and anomaly detection relies on relative distances to this representation rather than precise threshold tuning. We provide additional implementation details in the Appendix.

**Evaluation Criteria**: We use the Region-Based Detection Criterion (RBDC) and the Track-Based Detection Criterion (TBDC) as proposed in (Ramachandra & Jones, 2020) as our primary evaluation criteria and report the area under the curve (AUC) for false positive rates per frame from 0 to 1. We also report frame-level AUC (Mahadevan et al., 2010) scores. As highlighted in previous works (Ramachandra & Jones, 2020) frame-level AUC only evaluates temporal accuracy and disregards spatial localization of anomalies. RBDC and TBDC measure a method's capacity to accurately identify anomalous spatio-temporal regions within a given video sequence.

## 4.1 Quantitative Results

The results of our method on ComplexVAD, as well as those of EVAL (Singh et al., 2023), MemAE (Gong et al., 2019), AnomalyRuler (Wu et al., 2024c), and Scene-Graph Mumcu et al. (2025b), using the three evaluation criteria described above, are reported in Table 2. We also provide the result for the combination of our method and Scene-Graph on the ComplexVAD dataset. Our MLLM-EVAD method outperforms the next best method (Scene-Graph) by 5, 4 and 1 percentage points in RBDC, TBDC, and frame-level evaluations, respectively. We achieve the best result by combining the Scene-Graph method with ours, reaching 25%, 70%, and 63% in RBDC, TBDC, and frame-level evaluations, respectively. AnomalyRuler, a

| Method | RBDC | TBDC | Frame |
|---|---|---|---|
| Siamese AE (Ramachandra et al., 2020a) | 41.2 | 78.6 | 87.2 |
| Hybrid AE (Liu et al., 2021) | 41.1 | 86.2 | 91.1 |
| BG-Agnostic (Georgescu et al., 2021) | 65.1 | 66.9 | 92.3 |
| Hybrid AE (Liu et al., 2021) + PCAB (Ristea et al., 2021) | 62.3 | 89.3 | 90.9 |
| BG-Agnostic (Georgescu et al., 2021) + PCAB (Ristea et al., 2021) | 66.0 | 64.9 | **92.9** |
| SSMTL++v1 (Barbalau et al., 2023) | 40.9 | 82.1 | **93.7** |
| SSMTL++v2 (Barbalau et al., 2023) | 47.8 | 85.2 | 91.6 |
| AnomalyRuler (Wu et al., 2024c) | N/A | N/A | 89.7 |
| EVAL (Singh et al., 2023) | **68.2** | 87.6 | 86.0 |
| T-EVAL (Singh et al., 2024) | 67.5 | **89.7** | 88.0 |
| T-EVAL + MLLM-EVAD | **68.9** | **90.1** | 88.4 |

Table 4: Results on the CUHK Avenue dataset using the area under the curve (AUC) as a percentage for the RBDC and TBDC evaluation criteria for different methods. Bold face indicates the best score and blue indicates second best.

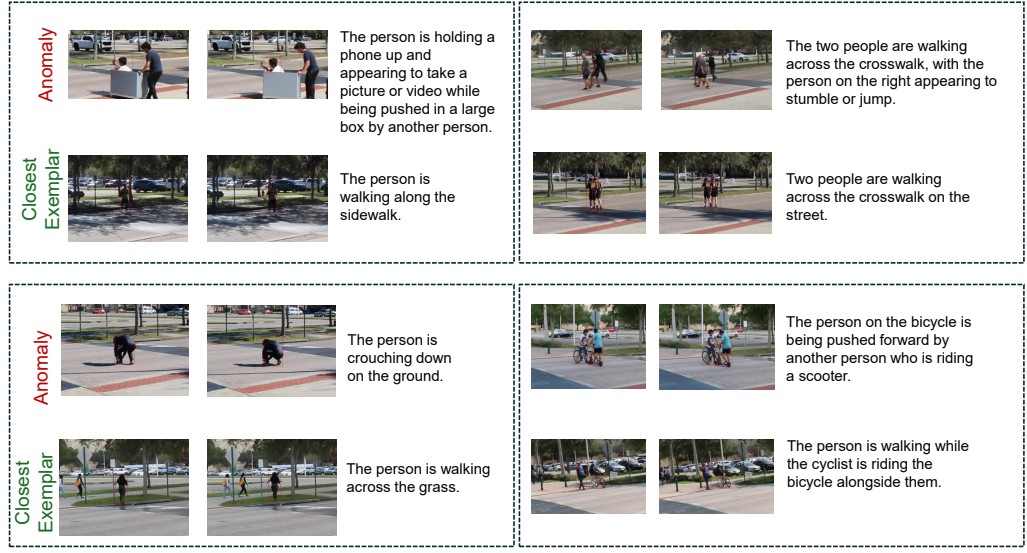

Figure 2: Examples of anomalies in the ComplexVAD dataset that are detected by our method. In each box, the top row shows two frames of the anomalous video clip followed by the textual description generated by Gemma 3. The second row shows two frames from the closest matching exemplar to the anomaly. The textual description for the closest exemplar is also shown. By contrasting the anomalous description with the closest normal description, it is clear why the testing video clip was found to be anomalous. The nominal description shown corresponds to the closest exemplar in the learned normal set and is presented for contrast; the anomaly is detected because its description is dissimilar from all nominal exemplars, and this semantic distance determines the anomaly score.

recent MLLM-based method (Wu et al., 2024c), achieves 54% frame-level AUC. However, we were not able to obtain results for the RBDC and TBDC criteria, as this method does not support spatial localization of anomalies. The MemAE method does very poorly for the two criteria that measure spatial localization. This implies that the regions of an image that MemAE predicts as anomalous are usually normal.

For the experiments on Avenue and Street Scene, we combine our method with the best-performing existing exemplar-based method, namely Tracklet EVAL (Singh et al., 2024). These datasets include many anomalies that need more fine-grained attributes (such as the direction and speed of travel for people and cars), so the high-level textual descriptions used by MLLM-EVAD are not sufficient on their own to detect all of

| Method | Explanation Quality (↑) |
|---|---|
| Human-written | $4.2 \pm 0.7$ |
| MLLM-EVAD | $3.8 \pm 1.1$ |

Table 5: Results of the human evaluation of explainability. Scores are reported as mean $\pm$ standard deviation of 5-point Likert ratings, aggregated across evaluators and anomaly clips.

the anomalies. We show that our textual descriptions do add an important new attribute, however, since, when combined with the more fine-grained attributes of Tracklet EVAL, the results with the MLLM textual descriptions improve accuracy for both datasets across all three evaluation criteria.

As shown in Table 4, on the Avenue dataset, the Tracklet-EVAL + MLLM-EVAD method improves the state-of-the-art for both the RBDC and TBDC criteria. On Street Scene (results shown in Table 3), the state-of-the-art is surpassed for the TBDC criterion. Since the frame-level criterion does not measure spatial localization accuracy which is the focus of these datasets, we would argue that the slightly-below SOTA frame-level numbers are not representative of the true accuracy of these methods. We should note that we were not able to exactly reproduce the results reported for Tracklet EVAL (Singh et al., 2024) on the Avenue dataset, most likely due to different parameters, so we report the results we could reproduce for that method and use that version in combination with MLLM-EVAD for the results on the combined method. All quantitative results reported in Tables 2, 4, and 3 are computed using raw, unsmoothed anomaly scores. Although temporal smoothing (e.g., Gaussian filtering) is commonly used for frame-level AUC evaluation, we do not apply smoothing in our main comparisons to ensure consistency across methods. For completeness, we additionally report Gaussian-smoothed frame-level results for our method on ComplexVAD in the appendix as a supplementary post-processing analysis.

## 4.2 Qualitative Results: Explainability

In addition to the quantitative results, we present qualitative results to demonstrate the explainability capabilities of our method. The combination of using an MLLM agent and an exemplar selection strategy enables intuitive explanations for detected anomalies by providing descriptions of both the anomaly and the most similar event from the nominal training video. In Figure 2, we demonstrate four detected anomalies from ComplexVAD along with their textual descriptions and the textual descriptions of the closest exemplars, as generated by our method. For example, one detected anomaly is described as "The person is crouching down on the ground." The closest matching exemplar from the nominal set is "The person is walking across the grass." The action of crouching deviates from the typical behavior observed in the training data. This difference highlights how our method leverages semantic understanding to identify uncommon activity, offering an interpretable explanation for the anomaly.

Another example involves an anomaly described as "The person is holding a phone up and appearing to take a picture or video while being pushed in a large box by another person." The closest matching exemplar from the nominal set is "The person is walking along the sidewalk.", which is, in fact, the most common activity involving a person in this dataset. The anomalous event, a person being pushed in a box, does not occur in the nominal video and represents an unusual and complex behavior. This example further demonstrates our method's ability to detect rare interaction patterns that deviate from typical activities in the scene. In addition, these examples illustrate how our method also provides reasonable explanations for detected anomalies. By leveraging descriptive comparisons with nominal exemplars, our method enables enhanced and explainable video anomaly detection.

As discussed in Section 5, the absence of ground-truth textual benchmarks makes quantitative evaluation of explainability difficult in the semi-supervised, single-scene VAD setting. To complement our qualitative analysis under this constraint, we perform a small-scale human evaluation of the generated explanations. We randomly select 10 anomalous video clips from ComplexVAD and, for each clip, consider two textual explanations: (i) a human-written annotation and (ii) the description generated by our system. For each clip, evaluators are shown the video and the two anonymized descriptions (labeled as "Description A" and "Description B") presented in randomized order, without revealing their source. Evaluators rate each

| Method | RBDC | TBDC | Frame |
|---|---|---|---|
| Sentence-BERT | 24.0 | 68.0 | 61.0 |
| BLEU | 17.0 | 69.0 | 58.0 |
| METEOR | 16.0 | 69.0 | 57.0 |

Table 6: The table reports results on the ComplexVAD datasets using the area under the curve (AUC) as a percentage for the RBDC, TBDC and Frame-Level evaluation criteria for the our method using 3 different sentence distance functions.

| MLLM | RBDC | TBDC | Frame |
|---|---|---|---|
| GPT-4o | 19.0 | 67.0 | 59.0 |
| Gemma 3 | 24.0 | 68.0 | 61.0 |

Table 7: The table reports the performance of our proposed method on the ComplexVAD datasets using the area under the curve (AUC) as a percentage for the RBDC, TBDC and Frame-Level evaluation criteria with different MLLM agents GPT-4o and Gemma 3.

description on a 5-point Likert scale according to the following question: "How well does this description explain why the event in the video is anomalous?" (1: not at all informative, 5: extremely informative). We collect ratings from 5 independent evaluators and report the mean and standard deviation of the scores aggregated across clips and evaluators in table 5. The results show that the explanations generated by MLLM-EVAD are consistently rated as informative and closely aligned with human-written explanations, supporting the interpretability claims of our approach. All human-written and MLLM-generated explanations used in this study are provided in Appendix D for qualitative inspection.

### 4.3 Ablation Study

We conduct two ablation studies by modifying the default settings of our method. In the first ablation study, instead of using sentence embeddings and cosine dissimilarity as the distance metric, we use the raw sentences and compute distances using BLEU (Papineni et al., 2002) and METEOR (Banerjee & Lavie, 2005). We emphasize that BLEU and METEOR are used here solely as alternative sentence similarity functions within the anomaly detection pipeline; no ground-truth textual annotations are involved, and evaluation is performed using standard VAD metrics. In Table 6, we compare the results of cosine distance over Sentence-BERT embeddings with those obtained using BLEU and METEOR, based on the descriptions generated by our method using GPT-4o as the MLLM agent. The results show that Sentence-BERT embeddings yield slightly higher accuracy in the RBDC and frame-level criteria while BLEU and METEOR perform slightly better on the TBDC criterion. Sentence-BERT is the preferred similarity function considering its greater efficiency due to embeddings which can be saved and only require a dot product for comparison.

In the second ablation study, we compare our method's performance on ComplexVAD using different MLLM agents, namely Gemma 3 and GPT-4o. In Table 7, we show that Gemma 3 achieves better performance, with 24%, 68%, and 61% in the RBDC, TBDC, and frame-level criteria, respectively, while GPT-4o yields 19%, 67%, and 59%. We attribute this difference to the fact that Gemma 3 generates more detailed and descriptive sentences compared to GPT-4o. This level of detail is particularly important for detecting interaction-based anomalies, where subtle contextual cues are crucial, and likely contributed to the observed performance gap. We provide a comparison of the descriptions generated by Gemma 3 and GPT-4o in the Appendix.

## 5 Limitations and Future Directions

While our framework shows promising results, we identify several limitations that pave the way for future research. First, our implementation relies on powerful but computationally expensive multimodal models like Gemma 3. Their high inference latency poses a challenge for real-time applications. Recent studies have shown that smaller, task-specific models can perform exceptionally well on specialized tasks (Wang et al.,

2020; Schick & Schütze, 2020; Marafioti et al., 2025; Mumcu & Yilmaz, 2024). We will explore fine-tuning such models for single scenes, specifically for describing object activities and interactions.

Second, a primary contribution of our method is its semantic explainability, but quantitatively evaluating this feature is difficult due to a lack of suitable benchmarks. Current datasets with textual ground truth are designed for multi-scene or weakly-supervised tasks and do not fit the semi-supervised, single-scene VAD problem (Zhu et al., 2024; Du et al., 2024; Zhang et al., 2025). Therefore, a key future direction is the development of new datasets with ground-truth textual descriptions for scene-specific anomalies.

Our framework relies on relative consistency across nominal textual descriptions rather than absolute textual correctness. Anomaly detection is performed by comparing sentence embeddings to a set of nominal exemplars, such that deviations from the learned nominal distribution determine anomaly scores. As a result, minor variations in wording or phrasing do not directly affect detection decisions, provided that nominal descriptions remain semantically consistent. In addition, we employ a fixed system prompt and a constrained task formulation that focuses on describing object activities and interactions, which further reduces sensitivity to prompt wording. Nevertheless, a more systematic evaluation of prompt sensitivity remains an important direction for future work, particularly given the computational cost of large-scale MLLM querying.

Finally, our framework currently leverages a robust traditional object detector, which is well-suited for scenarios with known object classes. To further generalize our approach, a promising future direction is the integration of open-vocabulary object detection methods (Liu et al., 2024; Cheng et al., 2024; Mumcu et al., 2025a). By enabling the recognition of an open set of object categories, this advancement would significantly broaden our system's applicability, allowing it to analyze and describe a more diverse range of anomalies in unconstrained, real-world environments.

## 6 Conclusions

In this work, we presented a novel video anomaly detection framework that leverages multimodal Large Language Models to detect and explain complex anomalies. By extracting object-level activity and interactions and converting them into textual descriptions, our method moves beyond traditional pixel level modeling and introduces a language-based, interpretable layer for video anomaly detection. Unlike many prior MLLM-based methods that rely on direct frame-level judgments, our approach builds a set of nominal textual descriptions and identifies anomalies through language-based comparison, offering both accuracy and explainability.

We demonstrate that our method outperforms existing approaches on ComplexVAD which is a challenging benchmark specifically designed to test interaction-based anomalies. In addition our method also improves of the state-of-the-art results on standard datasets like Avenue and Street Scene when combined with the fine-grained method Tracklet EVAL (Singh et al., 2024). Through ablation studies, we showed the importance of using sentence embeddings and detailed MLLM-generated descriptions. Notably, we found that Gemma 3 provided more informative descriptions than GPT-4o, contributing to higher detection accuracy.

Beyond strong quantitative results, our method offers clear and interpretable explanations for detected anomalies, highlighting its potential for real world deployment in safety critical surveillance applications. We believe this framework opens new directions for integrating multimodal large language models into video understanding tasks and sets the stage for future research in explainable and high level language-based video anomaly detection.

## 7 Broader Impact Statement

This work studies video anomaly detection with a focus on modeling scene-specific object interactions using multimodal large language models. While the proposed method is intended as a research contribution to representation learning and explainable anomaly detection, it is important to consider potential broader impacts and risks associated with its use.

A primary application domain of video anomaly detection is surveillance, which raises ethical concerns related to privacy, misuse, and surveillance overreach. If deployed irresponsibly, anomaly detection systems may contribute to unwarranted monitoring or automated enforcement. We emphasize that the proposed approach is intended as a decision-support tool rather than an automated decision-making system, and should be used in human-in-the-loop settings where outputs are interpreted by responsible operators rather than acted upon autonomously.

The use of multimodal large language models introduces additional considerations. MLLMs are computationally expensive, and converting large volumes of video data into textual descriptions increases energy consumption and carbon footprint compared to conventional VAD pipelines. For this reason, our method is not designed for real-time or large-scale deployment, and the MLLM component is used sparingly (offline and at sparse temporal intervals). Future work should consider more efficient models or distillation strategies to reduce environmental impact.

MLLM-generated textual descriptions may also hallucinate or include spurious details. In safety-critical scenarios, such hallucinations could lead to misleading explanations or mischaracterization of events. Our framework mitigates this risk by relying on relative consistency across nominal data and by using language-based representations as an auxiliary explanatory signal rather than as a sole decision criterion. Nevertheless, we caution against interpreting generated text as ground truth or using it without human oversight.

Finally, like many video analysis methods, our approach inherits biases present in training data and object detection models. Dataset bias may affect which behaviors are considered "normal" or "anomalous," potentially leading to unfair or incorrect outcomes if deployed without careful consideration. We encourage future work to examine dataset diversity, bias mitigation, and domain-specific validation prior to any real-world deployment.

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

## A    Compute Resources

All of our experiments were run on a cluster of eight 80 GB NVIDIA A100 GPUs. A single 80 GB GPU would be sufficient and multiple GPUs were used only to reduce the total compute time by processing multiple video sequences in parallel.

For experiments using the locally deployed MLLM, we report inference latency measurements using the Gemma model. On a single NVIDIA A100 GPU, querying Gemma with paired object-centric crops and the associated prompt takes on average 0.58 seconds per query, with a standard deviation of 0.16 seconds (measured over repeated runs). This provides a reproducible lower bound on the inference cost of the MLLM component under local deployment.

Our experiments that used GPT-4o accessed it through an API via the cloud.

## B    Cropping Frames

For each selected object pair $o_1$ and $o_2$, we first compute a single bounding box that tightly encloses both objects in a given frame. Let the individual bounding boxes for the two objects in frame $F_t$ be denoted as $\mathrm{bbox}_a = (x_{1a}, y_{1a}, x_{2a}, y_{2a})$ and $\mathrm{bbox}_b = (x_{1b}, y_{1b}, x_{2b}, y_{2b})$, where $(x_1, y_1)$ represents the top-left corner and $(x_2, y_2)$ the bottom-right corner of a bounding box. These are merged into a single bounding box $\mathrm{bbox}_1$ by taking the element-wise minimum for the top-left corner and maximum for the bottom-right corner:

$$
\begin{aligned}
\mathrm{bbox}_1 = \big( &\min(x_{1a}, x_{1b}),\ \min(y_{1a}, y_{1b}), \\
&\max(x_{2a}, x_{2b}),\ \max(y_{2a}, y_{2b}) \big)
\end{aligned}
\tag{3}
$$

We apply the same procedure to obtain $\mathrm{bbox}_2$ in the future frame $F_{t+\Delta}$, using the tracked locations of the same object pair.

Let $\mathrm{bbox}_1 = (x_{1,1}, y_{1,1}, x_{2,1}, y_{2,1})$ denote the merged bounding box of the selected object pair in frame $F_t$, and let $\mathrm{bbox}_2 = (x_{1,2}, y_{1,2}, x_{2,2}, y_{2,2})$ denote the merged bounding box of the same objects tracked in the future frame $F_{t+\Delta}$. A unified bounding box that covers both time steps is computed as:

$$
\begin{aligned}
\mathrm{bbox}_{\mathrm{merged}} = \big( &\min(x_{1,1}, x_{1,2}),\ \min(y_{1,1}, y_{1,2}), \\
&\max(x_{2,1}, x_{2,2}),\ \max(y_{2,1}, y_{2,2}) \big)
\end{aligned}
\tag{4}
$$

Let $(x_1, y_1)$ and $(x_2, y_2)$ represent the top-left and bottom-right corners of this merged bounding box. Let $w_{\min}$ and $h_{\min}$ denote half of the minimum desired crop width and height, respectively. In our case, we set $w_{\min} = 240$ and $h_{\min} = 135$, corresponding to a minimum crop size of $480 \times 270$ pixels. This size was chosen as a practical minimum that ensures sufficient context around objects while keeping computational overhead low. In our experiments, we found that a resolution of $480 \times 270$ served as a reliable lower bound, providing adequate spatial detail for capturing object activities without reducing the quality of the LLM generated descriptions. While larger crops may be used when available, we observed this setting to be a consistent and effective baseline across different datasets.

We then compute the crop dimensions as

$$
\begin{aligned}
w &= \max\!\left( \tfrac{|x_2 - x_1|}{2},\ w_{\min} \right), \\
h &= \max\!\left( \tfrac{|y_2 - y_1|}{2},\ h_{\min} \right)
\end{aligned}
\tag{5}
$$

The crop boundaries are determined as:

$$
\begin{array}{ll}
x_{\min} = \max(x_1 - w, 0), & y_{\min} = \max(y_1 - h, 0), \\
x_{\max} = \min(x_2 + w, W), & y_{\max} = \min(y_2 + h, H)
\end{array}
\tag{6}
$$

where $W$ and $H$ denotes the width and height of the full frame. The final cropped region $C_t$ is extracted from the current frame $F_t$ as:

$$C_t = F_t[y_{\min} : y_{\max},\, x_{\min} : x_{\max}]$$

To ensure both temporal frames share the same spatial context, we apply the exact same crop coordinates to the future frame $F_{t+\Delta}$:

$$C_{t+\Delta} = F_{t+\Delta}[y_{\min} : y_{\max},\, x_{\min} : x_{\max}]$$

This guarantees that both cropped frames contain the same background and object regions, allowing for consistent comparisons across time. In the case of a single object, the same procedure is applied with one exception: instead of merging two object bounding boxes in each frame, we directly use the bounding box of the single object in frame $F_t$ and its tracked location in $F_{t+\Delta}$. The rest of the cropping process, including the computation of the unified bounding box across time, minimum size enforcement, and cropping, is performed exactly as described above.

## C  Comparison of Gemma 3 and GPT-4o Activity Descriptions

In Figure 3, we share activity descriptions generated by Gemma 3 and GPT-4o for given images and the promt: *"Briefly describe what the <object name>s in the enclosed regions of these images are doing. The two images were taken one second apart."*

We have observed that Gemma 3 often gives more concise descriptions with fewer unnecessary details such as the color of a person's clothes. Moreover, Gemma 3 sometimes includes important details that GPT-4o omits, such as the fact that a person is using his or her phone.

| | Gemma 3 | GPT-4o |
|---|---|---|
|  | The person is crouching down on the ground. | The person in the enclosed region is bending over and appears to remain in the same position between the two images |
|  | The person is holding a phone up and appearing to take a picture or video while being pushed in a large box by another person. | In the images, the person standing seems to be pushing or interacting with a cart or box that the seated person appears to be inside or using. The standing person maintains their position or posture relative to the cart between the images. |
|  | The person on the bicycle is being pushed forward by another person who is riding a scooter. | The person in the blue shirt and the person on the bike are interacting, as the person in blue appears to be preventing the cyclist from moving forward. |
|  | The person is walking across the crosswalk while looking at their phone, and a cyclist is passing by them on the right. | The person in the enclosed region is stopping to allow a cyclist to pass by. |
|  | The two people are walking across the crosswalk, with the person on the right appearing to stumble or jump. | In the first image, the person in the black shorts and dark hair appears to be jumping or falling next to the bald person. In the second image, the person in the black shorts and dark hair is standing up straight, while the bald person remains standing. There is no apparent interaction between them. |

Figure 3: The activity descriptions generated by Gemma 3 and GPT-4o with given images.

# D    Human Evaluation Examples

| Clip ID | Human-written Explanation | MLLM-EVAD Explanation |
|---|---|---|
| 4201 | A car stops abruptly after the crosswalk, causing the car behind to halt on the crosswalk and block pedestrians before both continue. | The car is driving forward as a person with a backpack walks across the crosswalk in front of it. |
| 4254 | Two people cross the crosswalk when a third runs up and jumps onto one of them, who appears distressed. | The person in the black jacket appears to be pushing the person in the grey shirt, causing them to stumble forward. |
| 4288 | The skateboarder falls in the crosswalk as a scooter rider passes by. After a moment, the skateboarder gets up and leaves the crosswalk. | The person is running and then falls to the ground. |
| 4318 | A dog runs across the crosswalk without a leash, moving away from its owner. | The person was walking with the dog, but the dog has broken away and is now running ahead. |
| 4360 | A person hits another person on the head with a bat and the other person is visibly hurt. | The two people are walking across the crosswalk together. |
| 4371 | A scooter and a bicycle collide while crossing the crosswalk, then both riders continue through. | The person on the skateboard is being pulled forward by the person riding the bicycle. |
| 4405 | Two people play volleyball near the crosswalk, and their ball eventually falls toward the road. | The people in the enclosed region are playing with a ball, passing it back and forth to each other. |
| 4414 | A golf cart is parked near a crosswalk and then drives away. | The golf cart is driving forward as the person walks alongside it, and a car is passing in front of the golf cart. |
| 4426 | A scooter rider crosses a crosswalk, then stops, turns around, and rides back. | The person is riding the scooter forward across the pavement. |
| 4431 | A person climbs a pole near the crosswalk, sits briefly, then gets down and walks away. | The person is climbing onto a pole. |

Table 8: Examples used in the human explainability evaluation. For each anomalous clip, we show the human-written explanation and the explanation generated by MLLM-EVAD. Anomalous clips are randomly selected from ComplexVAD dataset.

# E    Effect of Temporal Gaussian Smoothing

| Method | Frame AUC (Raw) | Frame AUC (Gaussian Smoothed) |
|---|---|---|
| MLLM-EVAD | 0.61 | 0.68 |
| Scene-Graph + MLLM-EVAD | 0.63 | 0.71 |

Table 9: Effect of temporal Gaussian smoothing on frame-level anomaly detection performance on the ComplexVAD dataset. Smoothing is applied only as a post-processing step for this supplementary analysis and is not used in any main quantitative comparisons.

As stated in the main text, all quantitative results reported in Tables 2, 4, and 3 are computed using raw, unsmoothed anomaly scores to ensure consistency across methods.

As a supplementary analysis, we report the effect of applying Gaussian temporal smoothing to the frame-level anomaly scores of our best-performing configuration on the ComplexVAD dataset. Temporal smoothing is a commonly used post-processing step in video anomaly detection to reduce temporal noise in frame-level scores. We emphasize that smoothing is not part of the proposed method and is not used in any of the main quantitative comparisons; the results shown here are provided solely for completeness and illustrative purposes.

