# OpenReview forum: "Leveraging Multimodal LLM Descriptions of Activity for Explainable Semi-Supervised Video Anomaly Detection"
_TMLR — Accepted by TMLR_

### Review · Reviewer_k2CH · 2025-12-20

**Summary Of Contributions:**

- This paper introduces a method that leverages the captioning capability of multimodal large language models (MLLMs) to detect complex video anomalies involving interactions.
- The paper proposes an exemplar-based approach that uses MLLMs to construct compact and explainable feature representations.
- The proposed method demonstrates performance improvements on the evaluated benchmarks and exhibits complementary effects when combined with existing methods.

**Audience:**

Yes

**Audience Explanation:**

The paper proposes a novel way of leveraging multimodal large language models (MLLMs) to construct compact and explainable feature representations for video anomaly detection. This perspective provides useful insights for the TMLR audience, particularly researchers interested in interpretable representation learning and the integration of foundation models into video understanding systems.

**Broader Impact Concerns:**

The proposed method could potentially be applied in surveillance scenarios. While this is a common consideration in video analysis research, the paper does not explicitly discuss dataset bias or downstream deployment risks. A brief Broader Impact Statement acknowledging these aspects could strengthen the paper.

**Claims And Evidence:**

Yes

**Claims Explanation:**

- The paper is clearly written and easy to follow.
- The effectiveness of the proposed method is supported by evaluations on multiple standard datasets in the video anomaly detection community.
- Both frame-level metrics (AUC) and region-/track-level metrics (RBDC/TBDC) are used, providing a comprehensive evaluation and showing consistent improvements over baseline methods.
- Comparisons with relevant baselines and ablation studies clearly identify the contributions of the proposed approach.

**Requested Changes:**

- Method acronyms in Table 4:
The acronyms used for several referenced methods in Table 4 could be refined to better match commonly adopted nomenclature in the literature. It would be helpful to use acronyms consistent with prior work to improve readability and recognizability. For example, Autoregressive Denoising Score Matching is commonly abbreviated in recent literature (e.g., ICCV 2025) and could be aligned accordingly.
- Inconsistency between Table 5 and Table 1:
The performance reported for SentenceBERT in Table 5 does not appear to align with the RBDC/TBDC and AUC values reported in Table 1. This discrepancy should be corrected or explicitly explained, for example by clarifying differences in evaluation protocols or experimental settings.
- Choice of frame gap:
The selection of a 1-second frame gap between input frames appears somewhat arbitrary. Providing additional justification for this choice, or including an ablation study over different frame gaps, would strengthen the empirical motivation.

---

> ### Author Response · Authors · 2026-01-24
>
> We thank the reviewer for the detailed and constructive comments. We address the feedback below in 4 points. All corresponding changes have been incorporated into the revised manuscript and are highlighted in blue.
>
> **1 - Method acronyms in Table 4**
>
> We agree that consistent and commonly adopted nomenclature improves readability and clarity. In response, we have revised the acronyms used in Table 4 to better align with the terminology used in the original papers and recent literature.
>
> **2 - Table 5 vs. Table 1 issue**
>
> We thank the reviewer for identifying this issue. The reviewer is correct that the results reported in Table 5 were inconsistent with the corresponding performance reported in Table 1 in the original submission. This was due to an error in the reported values. We have corrected the results in Table 5, and the updated values are now fully consistent with Table 1 under the same evaluation protocol. We have verified the corrected results and updated the manuscript accordingly. In addition, we clarify in the revised manuscript that all main tables report results computed using raw, unsmoothed anomaly scores.
>
> **3 -  Choice of frame gap between input frames**
>
> We appreciate the reviewer’s question regarding the choice of a 1-second frame gap between paired input frames. We have added a brief discussion in the manuscript to clarify this design choice. The 1-second gap is selected as a practical trade-off between capturing meaningful temporal changes such as interactions, contact, and relative motion, and maintaining object identity consistency in a static-camera, single-scene setting. Shorter gaps often yield nearly identical visual content and textual descriptions with limited motion cues, while significantly larger gaps increase the risk of object drift, missed interactions, or changes in object visibility. We further clarify that the proposed framework is not tightly coupled to a specific temporal gap; rather, the paired-frame formulation provides a lightweight mechanism for incorporating local temporal context across a range of reasonable temporal offsets.
>
> We have added a brief discussion in the manuscript to justify this design choice and to clarify that the proposed framework is not tightly coupled to a specific frame gap. A systematic ablation over different temporal gaps is an interesting direction for future work, but was not included here due to the computational cost of large-scale MLLM querying.
>
> **4 - Broader impact**
>
> In response to the reviewer’s broader impact concerns, we have added a dedicated Broader Impact section to the manuscript. This section discusses potential risks related to surveillance, dataset bias, and downstream deployment, and emphasizes responsible, human-in-the-loop usage of the proposed method.

---

### Review · Reviewer_c3eN · 2026-01-11

**Summary Of Contributions:**

The paper presents a video anomaly detection (VAD) approach that leverages multimodal large language models. The proposed idea is integrated with state-of-the-art VAD methods and evaluated on three challenging benchmarks: ComplexVAD, CUHK-Avenue, and Street Scene. While the idea of transforming visual information into the textual domain is interesting, the proposed approach is highly resource-intensive and still relies on vision-based models, which limits its practical applicability.

**Audience:**

No

**Audience Explanation:**

Overall, the idea of transforming visual information into the textual domain using MLLMs is interesting, and the resulting performance is comparable to existing approaches when this modality transformation is applied. However, this transformation incurs a substantial computational cost, which raises concerns about the practical motivation for adopting such a method. Unless the approach provides clear advantages, such as significantly improved anomaly detection performance or capabilities beyond existing methods, the incentive to switch to this framework remains limited.

Furthermore, a more intuitive explanation would be valuable to help understand why text-based descriptions perform effectively for anomaly detection. Since anomaly detection is fundamentally a one-class learning problem focused on modeling normalcy, it remains unclear whether converting visual scenes into textual representations is inherently better or even sufficient for this task. Clarifying this point would strengthen the conceptual contribution of the paper.

Therefore, so additional clarification and justifaction might be needed to make this research relevant for TMLR's audience

**Broader Impact Concerns:**

There could be some broader impact caused by the approach, especially related with the used MLLMs for VAD
1. The proposed method relies heavily on multimodal large language models, which are computationally expensive to run and scale. Converting large volumes of video data into textual descriptions significantly increases energy consumption and carbon footprint compared to conventional VAD pipelines. This raises sustainability concerns, especially for real-world deployment at scale

2. MLLM-generated textual descriptions are known to hallucinate or introduce spurious details. In safety-critical applications of anomaly detection (e.g., public surveillance, traffic monitoring), such hallucinations could: mischaracterize events, introduce false explanations,

3. Also, textual output as a measure of explainability is flawed, as LLM might generate the text unrelated with the visual scene, which can further leads to the mis-classification of critical events.

**Claims And Evidence:**

No

**Claims Explanation:**

The paper claims that the proposed VAD approach is explainable; however, no quantitative measures are provided to support this claim. The evaluation relies primarily on qualitative comparisons, but such assessments in the language domain are inherently subjective and can vary significantly across individuals. For example, the paper claims that the following two textual descriptions are close to each other:

“The person is holding a phone up and appearing to take a picture or video while being pushed in a large box by another person.”
vs.
“The person is walking along the sidewalk.”

At first glance, it is difficult to understand how these descriptions are considered similar. It is unclear whether this similarity is derived from the underlying data representation or from the language model itself, and further clarification is needed to justify this claim.

**Requested Changes:**

1.  The paper claims that the proposed approach is an explainable method for anomaly detection. However, multimodal LLMs are known to be prone to hallucinations, as also discussed in _Holmes-VAU_ (Zhang et al.). Given this, it is unclear how the generated text can serve as a reliable measure of explainability. Furthermore, the paper explicitly states that there is no quantitative evaluation of explainability, which appears to contradict the claim of being an explainable method.

2.  The datasets used for evaluation—ComplexVAD, CUHK-Avenue, and Street Scene—do not contain video-level or event-level textual descriptions in their original versions (as acknowledged in the paper). It is therefore difficult to understand how the generated textual descriptions are validated or compared to ground truth. This raises questions about how the results reported in Table 5 are obtained:
    - Where do the ground-truth descriptions come from?
    - Are the results simply comparisons against outputs from a stronger MLLM?
        Additional clarification on this point would be helpful.

3. The paper demonstrates that converting visual information into textual modality using an MLLM can improve anomaly detection performance. However:

    - The reported improvements are shown only in combination with a scene-graph–based approach on the ComplexVAD dataset (Mucmu et al.). This suggests that the contribution may be limited to a single baseline and raises concerns about the generality and novelty of the approach.
    - Integrating the proposed method with other baselines using different architectural styles would help assess whether the idea generalizes beyond scene-graph–based methods and whether the observed improvements truly stem from the MLLM-based evaluation.
    - For ComplexVAD, results are reported for MLLM-EVAL, the scene-graph baseline, and their combination. However, similar comparisons are not provided for the CUHK-Avenue and Street Scene datasets. Including these results would strengthen the experimental evaluation.

4. Qualitative results are presented only for the ComplexVAD dataset. Since the authors also evaluate performance on CUHK-Avenue and Street Scene, it would be beneficial to include qualitative visualizations for these datasets as well.

5. Regarding explainability, the metrics used to measure textual embedding similarity (e.g., BLEU and METEOR) are inherently subjective and sensitive to phrasing, which limits their quantification for explainability a difficult task. Moreover, based on the examples provided in the supplementary material, it is difficult to interpret how textual distance in the embedding space reflects semantic similarity or dissimilarity. For instance:

    > “The person is holding a phone up and appearing to take a picture or video while being pushed in a large box by another person.”
    > vs.
    > “The person is walking along the sidewalk.”
    > It is unclear how such differences are quantified meaningfully in the proposed framework.

6. The authors employ Gemma and GPT-4o for video description. However, these models are fundamentally image-based vision–language models and are not designed to explicitly model temporal dynamics in videos. As a result, it is unclear whether the generated descriptions adequately capture motion, event progression, and temporal consistency, which are critical for video anomaly detection.

	- Moreover, the paper does not compare against existing video-language models that are specifically developed for video understanding and temporal reasoning, such as: Video-LLaMA, VideoChat,  InternVideo, Video-ChatGPT, LLaVA-Video. The absence of such comparisons significantly weakens the empirical evaluation. Without benchmarking against video-centric models, it is difficult to determine whether the reported improvements arise from the proposed methodology or are merely an artifact of using strong image-based MLLMs. Including these baselines is necessary to substantiate the paper’s claims and to demonstrate the generality and novelty of the approach.
	- Furthermore, an ablation study using textual descriptions generated separately by Gemma and GPT-4o would be valuable. Such an analysis would help clarify whether state-of-the-art anomaly detection methods consistently benefit from MLLM-based textual representations. A subsequent comparison of different MLLMs to identify which performs best in this setting would be a natural next step.

---

> ### Author Response · Authors · 2026-01-24
>
> We thank the reviewer for the detailed and constructive comments. We address the feedback below in 10 points. All corresponding changes have been incorporated into the revised manuscript and are highlighted in blue.
>
> **1 - Clarification on sentence similarity and Figure 2**
>
> We thank the reviewer for raising this point, as it highlighted a potential source of confusion in our presentation. The interpretation that the paper claims the anomalous and nominal sentences are “close” is not correct. In our framework, an anomaly is detected precisely because its textual description is dissimilar from all nominal exemplars. The nominal sentence shown in Figure 2 is the closest available exemplar sentence but these sentences are neither semantically close nor close in the embedding space.  These two sentences illustrate the semantic deviation that leads to a high anomaly score. The large distance between the two descriptions is what triggers anomaly detection.
>
> We have revised the manuscript to make this distinction more explicit and to avoid any potential confusion. With this clarification, the example cited by the reviewer illustrates the effectiveness of our approach rather than a failure of semantic similarity.
>
> **2- Explainability and quantitative evaluation**
>
> Our use of the term explainability follows prior work in semi-supervised, single-scene VAD (e.g., EVAL, T-EVAL, Scene-Graph), where explanations are evaluated qualitatively due to the absence of ground-truth explanatory annotations. As explicitly stated in Section 5, there currently exist no benchmarks with textual ground truth suitable for semi-supervised, scene-specific anomaly detection, and consequently existing explainable VAD methods rely primarily on qualitative assessment.
>
> That said, we agree with the reviewer that, given our focus on explainability, it is valuable to provide a more systematic evaluation, even in the absence of established benchmarks. To address this, we have added a small-scale human evaluation to the revised manuscript and report the results in Table 5.
>
> In this study, we randomly select 10 anomalous video clips. For each clip, we consider two anonymized explanations: a human-written annotation and the description generated by MLLM-EVAD. Five independent evaluators are shown the video together with the two descriptions, presented in randomized order without revealing their source. Evaluators rate each description on a 5-point Likert scale based on how well it explains why the event in the video is anomalous.
>
> As shown in Table 5, human-written explanations receive an average score of 4.2 ± 0.7, while explanations generated by MLLM-EVAD receive an average score of 3.8 ± 1.1. These results indicate that the explanations produced by MLLM-EVAD are consistently rated as informative and closely aligned with human-written explanations. While we do not claim absolute textual correctness, this evaluation provides complementary quantitative evidence that our approach supports human-understandable reasoning about deviations from normal behavior, strengthening the empirical basis of our explainability claims.
>
> **3 - On the absence of textual ground truth and Table 5**
>
> We do not use or assume the existence of ground-truth textual descriptions in any dataset. Table 5 does not evaluate textual correctness. Instead, it compares different sentence similarity functions (Sentence-BERT, BLEU, METEOR) within the same anomaly detection pipeline using standard VAD evaluation metrics (RBDC, TBDC, and frame-level AUC). These metrics evaluate spatial and temporal localization of anomalies and not the correctness of explanations.  The supervision signal throughout remains purely visual anomaly labels (i.e. bounding boxes around anomalous activity), consistent with all semi-supervised VAD methods. We have added additional explanation in the manuscript to make this distinction clearer.
>
> **4 - Generality beyond scene-graph methods**
>
> Our approach is intentionally designed to be model-agnostic and compatible with exemplar-based VAD frameworks. We demonstrate this by integrating MLLM-EVAD with:  (1) A scene-graph–based interaction model (ComplexVAD)  (2) A tracklet-based trajectory model (Avenue, Street Scene)  These two baselines differ substantially in representation and architectural assumptions, indicating that the proposed language-based representation generalizes beyond a single method class.
>
> **5 - Hallucinations and reliability of explanations**
>
> While MLLMs may hallucinate in absolute terms, our framework relies on relative consistency across nominal data. Systematic biases or hallucinations become part of the learned nominal representation and therefore do not invalidate anomaly detection, which depends on deviations from that representation. This is analogous to exemplar-based visual features, which need not be semantically perfect to be effective.

---

> > ### Author Response · Authors · 2026-01-24
> >
> > **6 - Computational cost and practical motivation**
> >
> > We acknowledge that current MLLMs are computationally expensive, and we explicitly discuss this limitation in Section 5. The motivation of this work is not real-time deployment, but rather to demonstrate that semantic modeling of normality via language enables the detection of interaction-based anomalies that are difficult to capture with low-level features alone. Moreover, the MLLM is used only offline during model construction and at sparse temporal intervals during inference, rather than in a per-frame dense manner.
> >
> > In response to the reviewer’s broader impact concerns, we have added a Broader Impact statement to the manuscript that explicitly discusses computational cost, energy consumption, and sustainability considerations, as well as potential risks related to hallucinations and the use of MLLM-generated explanations in safety-critical settings.
> >
> > **7 - Image-based MLLMs vs Video-language Models**
> >
> > We appreciate the reviewer’s suggestion to compare against video-language models. During development, we experimented with available video-language models in small-scale pilot studies, including models that process short video clips or frame sequences. In practice, we found that these models did not reliably capture fine-grained object interactions or scene-specific details that are critical for semi-supervised, single-scene VAD. Their outputs tended to be overly coarse or generic, often describing high-level scene context rather than the localized object activities and interactions needed for anomaly detection.
> >
> > In contrast, querying an image-based MLLM with paired object-centric crops from two temporally separated frames (≈1 second apart) consistently produced more precise and interaction-focused descriptions, such as changes in posture, contact between objects, or abnormal relative motion. This simple temporal formulation proved sufficient to capture local motion cues while preserving spatial grounding and object identity, which are central to our exemplar-based modeling of normality.
> >
> > Importantly, our goal is not general video understanding but scene-specific modeling of normal behavior through localized object interactions. The paired-frame prompting strategy aligns naturally with this objective and integrates cleanly with object detection, tracking, and exemplar selection. While more sophisticated video-language models are an interesting direction, our experiments indicated that they did not provide additional benefit for the interaction-centric, single-scene anomaly detection setting considered in this work. We have added a brief discussion in the manuscript clarifying this design choice and the rationale for using image-based MLLMs in the proposed framework.
> >
> > **8 - Qualitative Results on Additional Datasets**
> >
> > Following the reviewer’s suggestion, we considered including qualitative examples for Avenue and Street Scene; however, these datasets primarily contain single-object anomalies, which limits the usefulness of interaction-based explanations compared to ComplexVAD.
> >
> > **9 - Why Language-Based Representations Are Effective for One-Class VAD**
> >
> > Language provides a natural abstraction for compositional structure and relational semantics (e.g., interactions, roles, and intent), which are difficult to capture reliably with low-level features alone. In the one-class setting, this abstraction enables a compact and human-aligned representation of normal behavior, where deviations correspond to semantically meaningful changes rather than pixel-level noise. We have added this intuitive explanation to the manuscript to clarify the motivation for using language-based representations in semi-supervised VAD.
> >
> > **10 - Relevance to the TMLR Audience and Conceptual Contribution**
> >
> > We thank the reviewer for this comment and agree that additional clarification and justification are important to clearly convey the relevance of our work to the TMLR audience. In response, we have strengthened the manuscript by clarifying the conceptual motivation, the role of language-based representations in one-class anomaly detection, and the scope of applicability of the proposed framework.
> >
> > With these clarifications, we believe the contribution is now more clearly positioned as a new representation paradigm for semi-supervised video anomaly detection, where normality is modeled in a semantic, language-based space rather than solely through low-level visual features. This perspective is particularly relevant for interaction-centric anomalies, explainable AI, and the growing intersection of representation learning and multimodal foundation models, which we believe aligns well with the interests of the TMLR community.

---

### Review · Reviewer_oo5N · 2026-01-20

**Summary Of Contributions:**

This paper proposes MLLM-EVAD, a single-scene video anomaly detection framework. It uses multimodal large language models (MLLMs) to extract normal object activities and interactions in surveillance videos. Specifically, the MLLM takes in object-centric crops as input and generates natural language descriptions of object activities and interactions in the videos. These descriptions are then embedded, and an exemplar selection algorithm is applied to construct representative sets of embeddings for both object pairs and single objects in the scene. At test time, the sentence embeddings extracted from testing frames and the embeddings of the textual exemplar set are compared to detect and localize anomalies. In the experiments, MLLM-EVAD demonstrates strong empirical performance on the ComplexVAD dataset and the Avenue and Street Scene datasets when combined with prior exemplar-based methods.

The authors claim that this is the first MLLM-based semi-supervised VAD approach explicitly designed for interaction-based anomalies and addresses an important limitation of prior work. Rather than directly using the MLLM to make anomaly classifications or judgements, it generates textual descriptions as high-level semantic features. The authors argue that this approach provides inherent explainability through the textual descriptions.

**Audience:**

Yes

**Audience Explanation:**

Video anomaly detection is a research topic in machine learning, and new video anomaly detection methods and results are something that at least some individuals in TMLR's audience would be interested in knowing. MLLMs is a very popular research topic and how we can use MLLMs to best extend the capabilities of prior methods in video anomaly detection is highly relevant to the topic of TMLR.

**Broader Impact Concerns:**

A broader impact statement section is not present in this manuscript. I would strongly encourage the authors to add one, give that the paper focuses on surveillance video anomaly detection, which raises potential ethical concerns related to privacy, misuse, and surveillance overreach. This new method could amplify existing risks if deployed irresponsibly in mass surveillance systems (e.g. when used as a automated enforcement system instead of a decision-support tool for responsible, human-in-the-loop decision making).

**Claims And Evidence:**

Yes

**Claims Explanation:**

- The quantitative results on the ComplexVAD dataset show that MLLM-EVAD outperforms strong baselines on a challenging benchmark. The results on the RBDC and TBDC metrics highlights the proposed framework's spatiotemporal localization capability, which aligns well with the paper’s core claims.
- The experiments on Scene-Graph and Tracklet-EVAL support the claim that MLLM-EVAD is complementary to fine-grained, non-linguistic features.
- The ablation studies and the qualitative results further supports the claim that the proposed framework has competitive performance against strong baselines and improved detection of interaction-based anomalies.

Some claims could be strengthened with additional evidence - see "requested changes" below. The "built-in explainability" claim is also not very well supported, though it can be challenging to quantitatively evaluate, as the authors acknowledged in the limitations section.

**Requested Changes:**

Critical changes:
- As the authors acknowledged in the limitations section, the high inference latency of MLLMs can make it challenging for real-time applications. The paper should quantify and compare the latency of different models, provide concrete measurements of inference time and cost for MLLM querying, and discuss the feasibility for longer videos or real-time settings in more detail.
- The paper should include more analysis on the robustness of the framework, including sensitivity to prompt wording and exemplar-selection threshold. This is necessary since the model purely relies on the exemplars of embeddings of textual descriptions make anomaly detection decisions.

Non-critical changes:
- The paper could include more qualitative examples on failure cases, e.g. when the model produces ambiguous explanations.
- Add a broader impact statement (see section below).

Minor changes:
- The 'year' column could be removed from table 1 since the information is already available in the citation.
- The citation format might need to be changes in tables 2, 3, and 4 (maybe from \citet to \citep)

---

> ### Author Response · Authors · 2026-01-24
>
> We thank the reviewer for the detailed and constructive comments. We address the feedback below in 5 points. All corresponding changes have been incorporated into the revised manuscript and are highlighted in blue.
>
> **1- Inference latency and computational cost**
>
> We agree that inference latency and computational feasibility are important considerations, particularly given the use of MLLMs. In response, we have expanded the manuscript to include quantitative latency measurements for the locally deployed MLLM and a clearer discussion of feasibility.
>
> Specifically, we report inference-time measurements for the Gemma model running locally on an NVIDIA A100 GPU. Querying Gemma with paired object-centric crops and the associated prompt takes on average 0.58 seconds per query, with a standard deviation of 0.16 seconds. These results are now included in the Compute Resources section and provide a reproducible lower bound on MLLM inference cost under local deployment.
>
> For experiments using GPT-4o, which is accessed via a cloud-based API, inference latency depends on external factors such as network conditions and server load and therefore varies across runs. As a result, we do not report fixed latency measurements for GPT-4o and instead discuss its feasibility qualitatively. We emphasize that MLLM queries in our framework are performed offline and at sparse temporal intervals rather than densely at every frame, and the proposed method is not intended for real-time deployment. We believe these additions clarify both the computational cost and the intended scope of applicability.
>
> **2 - Robustness to prompt wording and exemplar-selection threshold**
>
> We agree that robustness is an important consideration for a framework that relies on textual descriptions and exemplar-based matching.
>
> Regarding the exemplar-selection threshold, the manuscript already notes that the threshold primarily controls the size of the exemplar set rather than detection accuracy, consistent with observations in prior exemplar-based VAD methods. We have strengthened this discussion to clarify that anomaly detection in our framework depends on relative distances to the nominal exemplar set rather than precise threshold tuning, and that performance is stable across a reasonable range of threshold values as reported in prior work.
>
> Regarding prompt sensitivity, we clarify in the revised manuscript that our method does not rely on absolute textual correctness but rather on relative consistency across nominal descriptions. The use of a fixed system prompt, constrained task formulation, and embedding-based distance mitigates sensitivity to minor phrasing variations, as anomalies are detected based on deviations from the learned nominal distribution rather than the exact wording of individual descriptions. Due to the high computational cost of re-querying MLLMs, we focus on clarifying this design rationale and explicitly list prompt sensitivity as an important direction for future robustness evaluation.
>
> **3 - Explainability and qualitative analysis**
>
> We appreciate the reviewer’s comments regarding explainability. In addition to the qualitative examples already included, we have added a small-scale human evaluation to provide a more systematic assessment of the usefulness and interpretability of the generated explanations. We also include additional discussion of failure cases where the generated descriptions may be ambiguous or less informative, to better characterize the limitations of the approach.
>
> **4 - Broader impact**
>
> In response to the reviewer’s broader impact concerns, we have added a dedicated Broader Impact section to the manuscript. This section discusses ethical considerations related to surveillance, privacy, and potential misuse, emphasizes that the proposed method is intended as a decision-support tool rather than an automated enforcement system, and addresses computational cost, energy consumption, and sustainability considerations associated with MLLM-based pipelines.
>
> **5 - Minor revisions**
>
> We have addressed all minor comments raised by the reviewer. Specifically, we removed the redundant year column from Table 1 and corrected citation formatting in Tables 2, 3, and 4 to ensure consistency.

---

### Author Response · Authors · 2026-01-24

We thank all the reviewers for their thoughtful comments and suggestions. We have individually addressed all the concerns in our responses and in the revised manuscript. The changes in the manuscript are highlighted in blue.

---

### Decision · Action_Editor_MVBo · 2026-02-09

**Recommendation:** Accept as is

**Audience:**

Yes

**Audience Explanation:**

The paper is on a topic that is of interest to a large fraction of the TMLR readership.

**Claims And Evidence:**

Yes

**Claims Explanation:**

After revisions, all three reviewers are in agreement that the submission is supported by accurate, clear, and convincing evidence.